# The Emericellipsins A–E from an Alkalophilic Fungus *Emericellopsis alkalina* Show Potent Activity against Multidrug-Resistant Pathogenic Fungi

**DOI:** 10.3390/jof7020153

**Published:** 2021-02-21

**Authors:** Anastasia E. Kuvarina, Irina A. Gavryushina, Alexander B. Kulko, Igor A. Ivanov, Eugene A. Rogozhin, Marina L. Georgieva, Vera S. Sadykova

**Affiliations:** 1Laboratory for Taxonomic Study and Collection of Cultures of Microorganisms, Gause Institute of New Antibiotics, st. Bolshaya Pirogovskaya, 11, 119021 Moscow, Russia; nastena.lysenko@mail.ru (A.E.K.); irina-alekcandrovna2013@yandex.ru (I.A.G.); i-marina@yandex.ru (M.L.G.); 2Clinical Antituberculosis Center, Moscow Government Health Department Scientific, st. Stromynka, 10, 107014 Moscow, Russia; kulko-fungi@yandex.ru; 3Laboratory of Neuroreceptors and Neuroregulators, Shemyakin and Ovchinnikov Institute of Bioorganic Chemistry, RAS, st. Miklukho-Maklaya, 16/10, 117997 Moscow, Russia; chai.mail0@gmail.com; 4Department of Biology, Lomonosov Moscow State University, Leninskie Gory 1-12, 119234 Moscow, Russia

**Keywords:** alkalophilic fungi, antifungal peptide, *Emericellopsis alkalina*, emericellipsins A–E, cytotoxic activity

## Abstract

Novel antimicrobial peptides with antifungal and cytotoxic activity were derived from the alkalophilic fungus *Emericellopsis alkalina* VKPM F1428. We previously reported that this strain produced emericellipsin A (EmiA), which has strong antifungal and cytotoxic properties. Further analyses of the metabolites obtained under a special alkaline medium resulted in the isolation of four new homologous (Emi B–E). In this work, we report the complete primary structure and detailed biological activity for the newly synthesized nonribosomal antimicrobial peptides called emericellipsins B–E. The inhibitory activity of themajor compound, EmiA, against drug-resistant pathogenic fungi was similar to that of amphotericin B (AmpB). At the same time, EmiA had no hemolytic activity towards human erythrocytes. In addition, EmiA demonstrated low cytotoxic activity towards the normal HPF line, but possessed cancer selectivity to the K-562 and HCT-116 cell lines. Emericillipsins from the alkalophilic fungus *Emericellopsis alkaline* are promising treatment alternatives to licensed antifungal drugs for invasive mycosis therapy, especially for multidrug-resistant aspergillosis and cryptococcosis.

## 1. Introduction

Infections acquired in healthcare settings are among the major causes of increased morbidity and mortality among hospitalized patients. There is a real and significant threat to human health as a result of infections caused by pathogenic fungi which are resistant to commonly used antifungal drugs. Current forecasts predict that broad-scale antimicrobial ineffectiveness is imminent [1,2,3] (http://www.life-worldwide.org/, accessed on 27 December 2020). Therefore, there is an urgent need to discover novel antifungal compounds with antimicrobial activity and to understand their mode of action [4,5].

Antimicrobial peptides (AMPs) are currently considered a promising alternative to antifungal drugs. Some naturally occurring and synthetically obtained AMPs have already been studied in clinical trials for the development of topical drugs to treat yeast infections of the skin [6,7]. The majority of natural AMPs are of mammalian origin (~75%), followed by plant (~13%) and bacterial origin (~10%). Only 1% of the currently known and studied AMPs are from fungi [8].

The bioactive AMPs produced by fungi include peptaibols, which are nonribosomal short peptides comprising around 2‒28 amino acids with both a peptide core and a nonpeptidic moiety. Peptaibols have demonstrated interesting bioactivities, including antibacterial, antitumor, and antiviral effects, as well as cytotoxicity and modulatory activity on mammalian glutamate-gated ion channels [9,10,11,12,13,14]. The fungi of the genus *Emericellopsis* are well known for their production of a peptaiboles with antibacterial and antifungal activity: The antiamoebins I‒XI from *Emericellopsis salmosynnemata* and *E.synnematicola*, the bergofungins A‒D from *E. donezkii*, the emerimicins II, III, and IV from *E. microspora* and *E. minima*, the heptaibin from *Emericellopsis* sp., the zervamicines from *E. salmosynnemata* (32 peptides grouped into five families according to the Norine databases; https://bioinfo.cristal.univ-lille.fr/norine/index.jsp, accessed on 27 December 2020).

The genus *Emericellopsis* (Hypocreales, Ascomycota) comprises 20 species as listed in the Index Fungorum (http://www.indexfungorum.org/, accessed on 27 December 2020) and MycoBank databases (https://www.mycobank.org/, accessed on 27 December 2020). It is noted that these fungi have a wide ecological amplitude and worldwide distribution [15,16,17,18]. Zuccaro et al. proposed creating two separate groups within *Emericellopsis* with different phylogeny and ecology, namely marine and terrestrial clades [16]. A new species of this genus (*E. alkalina*) was isolated from halophilic environments in 2013 [15]. The isolates grew well in a wide pH range (from 4.0 to 11.2) but with an optimum pH of 9‒11, showing an alkaliphilic phenotype. Grum-Grzhimailo et al. proposed another clade to include highly alkaline isolates of *E. alkalina*. Both growth patterns and molecular data suggest that this group is linked to the marine clade and may be of marine origin [15]. Natural products produced by alkalophilic fungi found in unusual and poorly studied ecosystems, such as soda lake sediments, represent a promising source of new valuable drugs.

Previously, the 65 strains of *E. alkalina* were isolated from soils near soda lakes in Siberia, Trans-Baikal regions (Russia), and Eastern Mongolia. Primary screening among isolates of this species revealed in *E. alkalina* strain VKPM F1428 the presence of antimicrobial activity against pathogenic micromycetes [19,20]. Furthermore, researchers carried out the isolation and structural and functional characterization of a new, previously undescribed, secreted antimicrobial peptide called emericellipsin A (EmiA), which is a product of nonribosomal synthesis and belongs to the group of peptaiboles [21].

The aim of this study was to characterize antimicrobial peptides called emericellipsins. A total of 65 ethyl acetate crude extracts of *E. alkalina* strains were investigated for antifungal activity and the presence of an emericellipsin complex. Structural elucidation of purified novel compounds called emericellipsins B–E with reference to fungal peptaibols was done using different analytical techniques, and the biological activities towards eukaryotic microorganisms were evaluated.

## 2. Materials and Methods

### 2.1. Isolation and Characterization of Alkalophilic Strains of Emericellopsis alkalina

All the strains were isolated from soils adjacent to soda or saline lakes (Figure 1A) and all of them showed the growth ability at pH 10.5 media: Alkaline agar (Figure 1B) and liquid alkaline medium (Figure 1C). The scanning electron microscopy (SEM) was used for the morphological study of the strains (Figure 1D–F). Isolates were characterized also based on multilocus DNA sequence analyses, i.e., large subunit rDNA, internal transcribed spacers 1and 2, including 5.8S rDNA, RPB2, TEF1-αand β-tub.

The definitions of the taxonomic status of the strains made during the description of the species confirmed that all of the strains used in this work belonged to the species *E. alkalina* Bilanenko and Georgieva (NCBI: txid1419734). Some of the isolates were deposited at the CBS-KNAW Fungal Biodiversity Center (Utrecht, The Netherlands), the All-Russian Collection of Microorganisms (VKM, Pushchino, Russia), and the Russian National Collection of Industrial Microorganisms (VKPM) [15].

### 2.2. Cultivation of the Fungi and Extraction of Emericellipsins A–E

The fungi were cultivated according to the previous protocol on a special alkaline medium at 26 °C in Erlenmeyer flasks under stationary conditions for 14 days [21]. The culture fluid (CF) was separated by filtration through membrane filters on a Seitz funnel under a vacuum. To isolate the antibiotic substances, the CF of the producers was extracted three times with ethyl acetate in an organic solvent/CF ratio of 1:5. The obtained extracts were evaporated in a vacuum on a Rotavapor rotary evaporator (Buchi, Switzerland) to dryness at 42 °C, the residue was dissolved in aqueous 70% ethanol, and the alcohol concentrates were obtained.

### 2.3. Purification and Identification of Emericellipsins A–E

#### 2.3.1. HPLC Analysis

The collected fractions were analyzed via LC-MS. The chromatographic separation was carried out using a Phenomenex Jupiter C4 column (150 mm × 2 mm, 2 µm) (Phenomenex, Torrans, CA, USA) in a linear gradient of acetonitrile in water from 5% to 70% with the addition of 0.1% formic acid. The MS acquisition was performed in a data-dependent MS2 mode. Collected HCD and CID spectra of peptaibols were merged and manually analyzed against the known structure of EmiA. The merged MS2 spectrum of EmiA showed an almost complete fragmentation pattern (b/y series), although the y series presented a relatively small intensity. Thus, this facilitated the mutation identification in the series of peptaibols. The mutation assignment was performed in a homologous manner, therefore, it only proposed sequences.

#### 2.3.2. MALDI TOF/TOF MS

The molecular masses of the individual fractions collected were measured by a matrix-assisted laser desorption/ionization (MALDI) time-of-flight (TOF/TOF) mass spectrometry, on an Autospeed MALDI-TOF instrument (Bruker Daltonics, Bremen, Germany), in a positive ion mode. We applied 2,5-dihydroxybenzoic acid (Sigma-Aldrich, Ronkonkoma, NY, USA) as a matrix for calibration. Mass spectra were analyzed with the FlexAnalysis version 3.4. software (Bruker Daltonics, Bremen, Germany).

#### 2.3.3. LC/ESI-MS Analysis

The purified compounds were analyzed by analytical ultra-performance liquid chromatography/mass spectrometry (UPLC-MS) using a Thermo Finnigan LCQ Deca XP Plus ion trap instrument with a Thermo Accela UPLC system (Thermo Fisher Scientific, Waltham, MA, USA). The samples were automatically applied on a YMC Triart microcolumn (C_18_ 150 mm × 2 mm, 1.9 µm) (YMC Co., Kyoto, Japan). Detection of absorbance was monitored by an UV/Vis diode array detector (UV-VIS DAD) (190–600 nm) and full scan mass spectrometry (MS) (electro spray ionization (ESI+), 150–2000 au). The fractions were predominantly dissolved in a mixture of water/methanol/acetic acid (88:10:2) up to a final concentration of 1 mg/mL, then filtered through a 0.45-µm nylon filter, and injected into the liquid chromatography (LC) system via the Auto Sampler equipment.

#### 2.3.4. UV Spectra

UV spectra of purified compounds were measured on a Shimadzu UV-1800 (Shimadzu Corp., Kyoto, Japan) with a scan range of 200‒340 nm.

### 2.4. Biological Assays

#### 2.4.1. Antifungal Activity

The antifungal activity of a crude peptaibols extract was measured by the disc-diffusion method. Discs of 6 mm in diameter containing 40 µL of sample were deposited on PDA agar plates (Sigma-Aldrich, St. Louis, MO, USA). The diameter of the inhibition zones was measured after 24h at 28 °C. The amphotericin B solution (Sigma-Aldrich, St. Louis, MO, USA) was used as a positive control. The minimal inhibitory concentration (MIC) value of each individual compound was determined using the broth two-fold microdilution method according to CLSI/NCCLS documents M27-A3, M38-A, and M38-A2 [22,23]. Yeast strains *Candida albicans* ATCC 14053 and the fungal strain *Aspergillus niger* ATCC 16404 were obtained from the American Type Culture Collection (ATCC, Manassas, VA, USA). *Aspergillus fumigatus* F-37 and *A. terreus* 3K were from the All-Russia Collection of Microorganisms (VKM, Pushchino, Russia). The spectrum of the antimycotic action of peptaibols was also evaluated in clinical isolates of filamentous fungi and yeast with multiple resistance. These were obtained from the collection of the Moscow Government Health Department Scientific and Clinical Antituberculosis Center (Moscow, Russia). *Aspergillus niger* 1133 m, *A. terreus* 497, *A. fumigatus* 390m, *C. albicans* 1582, *C. glabrata* 1402, *C. tropicalis* 156, *C. krusei* 1447, *C. parapsilosis* 571, *Cryptococcus neoformans* 297, and *Cr. laurentii* 325m were isolated from patients with invasive pulmonary aspergillosis and oropharyngeal HIV-positive patients with cryptococcosis. All clinical isolates demonstrated azole resistance to commercial fluconazole in vitro (Appendix A).

The RPMI 1640 medium with L-glutamine and phenol red, without sodium bicarbonate (ICN Biomedicals, Irvine, CA, USA), pH 7.0, was used in the study. New compounds were dissolved in dimethyl sulfoxide (DMSO) at 1600 μg mL^−1^. Serial dilutions (down to 3.13 μg mL^−1^) were prepared from stock solutions in the same solvent, then diluted 50-fold in the test medium and then twice when inoculated with the microbial suspension before incubation. The final solvent concentration was 1%. MICs were measured after cultivation at 35 °C for 24 h for yeast and 48 h for fungi as the lowest concentrations of agents that prevent any visible growth. To determine the minimal fungicidal concentration (MFC), a sample (100 µL) identified as having the MIC and the wells with the other three higher concentrations were plated onto the PDA agar. The lowest concentration causing 95% death of the fungal population was considered to be the MFC.

#### 2.4.2. Cell Culture and Cytotoxicity Assays

Experiments for determining the cytotoxic activity of the compound in MTT assays were performed as previously described [24]. The following tumor cell lines were used for this study: HCT-116 (colon cancer cell line), B16 (mouse melanoma cell line), K-562 (leukemia cell line), MDA-MB-231 (breast cancer line), and MCF-7 (breast cancer cell). Human postnatal fibroblasts were used as a normal cell line, and doxorubicin was used as a positive control. The cell lines were cultured in Dulbecco modified Eagle’s medium supplemented with 5% fetal calf serum, 2 mM Lglutamine, 100 U/mL penicillin, and 100 mg/mL streptomycin at 37 °C, 5% CO_2_ in a humidified atmosphere. Experiments were carried out on cells in the logarithmic phase of growth. The peptide was dissolved in DMSO as a 10 mM stock solution, followed by serial dilutions in water immediately before the experiment. Briefly, cells (5 × 10^3^ in 190 mL of culture medium) were plated in a 96-well plate and treated with 0.1% DMSO (blank control) or with 10 mM of the peptide (0.10‒50 μM; each concentration in duplicate) for 72 h. After incubation with peptide, 20 mL of aqueous MTT solution (3-(4,5-dimethylthiazol-2-yl)-2,5-diphenyltetrazolium bromide, 5 mg/mL) were added into each well for 2‒3 h. Formazan was dissolved in DMSO, and the absorbance at 570 nm was measured. The cytotoxicity at a given EmiA concentration was calculated as the percentage of absorbance in wells with peptide-treated cells compared to that of blank control cells (100%). The IC_50_ (50% growth inhibitory concentration) was defined as the concentration of EmiA that inhibited MTT conversion by 50%. All the experiments were repeated three times, each time in triplicate. EmiA was tested at a range of concentrations, 0.1‒10 μM.

#### 2.4.3. Hemolytic Activity

Erythrocytes were isolated from the peripheral blood of healthy donors after incubation at 4 °C for 2–3 h. All the donors are males, 35–38 years old. All of them have approved to collect blood samples, and the authors have permissions filled. All the blood samples have been destroyed after the experiments finished. The red blood cell suspension (100 μL) was pelleted, washed twice with saline, then the pellet was resuspended in 500 μL saline or distilled water (positive control of hemolysis). The compounds were added (final concentration of 20 μM). The samples were incubated for 1 h at 37 °C and centrifuged at 2000 rpm for 1.5 min. The OD of supernatants was measured on a Multiscan spectrophotometer FC (Thermo Scientific, Waltham, MA, USA) at a wavelength of 540 nm. The percent hemolysis was calculated relative to the positive control, taken as 100%.

Erythrocytes were isolated from the human blood: On the day of the study, blood was taken from the donor’s cubital vein into a test tube with an anticoagulant. The blood was incubated at 4 °C for 2‒3 h. A suspension of erythrocytes (100 μL) was diluted with a physiological buffer (pH 7.2) to a total volume of 500 μL. Solutions of the test substances with an initial concentration of 10 μM (DMSO) were diluted with a PBS buffer (1:10). The resulting solutions were introduced into Eppendorf tubes in the amounts required to create the studied concentration (5, 10, or 20 μM). A mixture of PBS with erythrocytes was added to a total volume of 200 μL. The control (C) was a mixture of erythrocytes with H_2_O (100% of hemolysis). The intact control solution (I) was a mixture of erythrocytes with the PBS and solvent (DMSO). The antibiotic gramicidin S (Gram S) was used as a control sample with active hemolysis. The mixture was incubated for 1 h at 37 °C. The measurement of the optical density of the supernatants was carried out on a BioTek ELx800 automatic microplate photometer (USA) at 570 nm. The optical density of the liquid above the sediment in the control sample with H_2_O was taken as 100%.

#### 2.4.4. Heterochromatin Condensation in Human Buccal Epithelium Cells

The quantitative analysis of heterochromatin granules located inthe nucleiofhumanbuccal epithelium cells was performed as previously described [25]. EmiA was applied at concentrations ranging from 25.0 to 400 µM.Donors A, B, and C are represented by three different humans who provided a donor tissue. All the donors are non-smoking, male and female, 20–22 years old. All of them have approved to collect their epithelial tissues, and the authors have permissions filled. All the tissue samples have been destroyed after the experiments finished.

## 3. Results

The antagonistic ability of the alkalophilic fungi *E. alkalina* was determined by producing antifungal compounds to inhibit the test pathogen. Among the 65 strains, 17 (accounting for 26.15%) had significant inhibitory activity against *Aspergillus niger* ATCC 16404. In some variants, an antifungal effect was also observed for *Candida albicans* ATCC 14053 (Table 1).

### 3.1. Antimicrobial Activity of Crude Extracts of E. alkaline Strains

After an investigation of all the strains from the *E. alkalina* collection, 17 fungal cultures producing emericellipsins were selected, and the qualitative peptaibol profiles of the strains were apparent. Compound EmiA was produced by most of the active strains (14 of the 17 examined), whereas other homologues were produced only by certain strains. Interestingly, the test of antifungal activity against *Candida* depended on the presence of a major component of EmiA in a total peptaibol extract (Table 1). According to their peptaibol profiles, the strains producing only homologues B–E were inactive towards *Candida albicans* ATCC 14053 in agar diffusion assays.

### 3.2. Isolation of the Individual Fractions from E. alkalina Concentrate

We applied an approach based on an organic extraction of *E. alkaline* VKPM F1428 liquid culture followed by a two-step fractionation of the concentrate obtained using a liquid chromatography technique. As a result, after analytical reversed-phase HPLC, a typical profile was obtained, which contains a number of peaks eluted from the column from 60.0 to 66.5 min. One of them was found to be emericellipsin A with a retention time of 63.7 min. The other dominant peaks were collected manually to provide the initial structural analysis (Figure 2). The total yield calculated for all these fractions was about 2.5% relative to the weight of the whole concentrate (2.5 mg per 100 mg); the major compound, EmiA, had the highest concentration and accounted for about 1.2 mg per 100 mg of dry weight.

### 3.3. Initial Structural Analysis of the Individual Fractions

The analytical reverse-phase HPLC was used to clarify the degree of molecular diversity of fungal metabolites which are chemically close to peptaibols. Compounds that are hydrophobic to different degrees (based on the retention time in separation by liquid chromatography) were found in the elution zone of the main component, emericellipsin A. Previously, it was shown that, among the entire spectrum of metabolites of *E. alkalina* A118, two compounds in addition to substance A118/37 (emericellipsin A) possessed similar functional activity: A118/35 (B) and A118/36 (C) [19]. These two compounds were eluted from the column with lower retention times (61.4 and 62.9 min, respectively) and displayed strong antimicrobial activity which was close to that of emericellipsin A. In the present study, we attempted to identify all the complexes of the metabolites eluted in the zone of emericellipsin A elution (Figure 1).

We were able to visualize a set of peaks eluted from the column in a short time range, and they were preliminarily classified as emericellipsin-like peptides based on their total UV spectra (data not shown). Therefore, we supposed that they had structural similarity to the main component. Taking this into account, we individually collected two more components with retention times of 65.1 (D) and 65.7 min (E).

### 3.4. Identification of the Active Fractions and Structure Diversity of Peptaibols from E. alkalina

According to previously obtained data, the structure of the main antibiotic, emericellipsin A as determined by hetero nuclear NMR spectroscopy, has the following form: Methyldecanoyl-MePro-AHMOD-Ala-Aib-Ile-Iva-bAla-Alaol-Glyol. Furthermore, it contains nine amino acid residues with modification of the N-terminal amino group and C-terminal hydroxyl [21]. Accordingly, based on the reference structure, fragmentation was carried out on a mixture of compounds A, B, C, D, and E, together with the original emericellipsin A, by LC-ESI mass spectrometry.

The results made it possible to identify all four compounds as close homologues of emericellipsin A with single substitutions: From isovaline to α-aminoisobutiric acid (“B” form), from alanine to serine at position 3 (“C” form), from α-aminoisobutiric acid to isovaline at position 4 (“D” form), and from alaninol to α-aminoisobutiric at position 8 (“E” form). All the identified molecules belong to the peptaibol group and were called emericellipsins B–E. Figure 3 shows the molecular ion masses of the identified compounds, and Table 2 presents the complete amino acid sequences. All the MS/MS spectra measured for EmiA–E are located in Appendix A.

### 3.5. Antifungal Activity of Emericellipsins A–E by Disc-diffusion Assay

EmiA–E were tested for antifungal activity on opportunistic and clinical fungal isolates. Major differences were observed in the activity of EmiA and homologues in all molds (Table 3). The lead compound EmiA inhibits the whole panel of opportunistic and clinical fungal isolates in agar dilution assays. EmiD and EmiE were totally inactive against all reference and clinical strains of *Aspergillus*. EmiB and EmiC displayed low activity only against *A. niger* strains, whereas both of these peptides were inactive against another *Aspergillus* spp. After incubation, the inhibition zones for clinical isolates of *Candida* and *Aspergillus* were found to be 20–25 and 15–19 mm for EmiA, while they were 10–12 and 15–17 mm for AmpB, respectively (Appendix A).

The MIC values for EmiA were in the 0.5–4 µg/mL range and confirmed the higher activity against both opportunistic and clinical isolates (Table 4 and Appendix A). The MIC of EmiB for most of the isolates of *Aspergillus* spp. ranged from 4 to 8 µg/mL. In contrast to EmiA, most isolates of *Aspergillus* spp. were insensitive to EmiD (MIC range: 16–32 µg/mL). No activity was observed against the test strains for EmiE (MIC > 64 µg/mL). The *A. terreus* isolates were also insensitive to AmpB.

The lead compound, EmiA, was selected for further functional analysis due to its strong antifungal activity and high amount in the crude extract.

### 3.6. Activity of Emericellipsin A against an Extended Panel of Yeast Clinical Isolates

To investigate the fungal activity of EmiA in more detail, we tested it against an extended panel of clinical yeast isolates including *C. albicans*, *C. glabrata*, *C. krusei*, *C. tropicalis*, *C. parapsilosis*, *Cryptococcus neoformans*, and *Cr. laurentii*. We determined the MIC for caspofungin, itraconazole, voriconazole, and amphotericin B (frequently used commercial antifungal drugs) for all of the tested clinical isolates in parallel with the AMP testing in order to confirm the resistance phenotype. The isolates *Cryptococcus neoformans* and *Cr. laurentii* were selected based on their resistance phenotype, including resistance to caspofungin, micafungin, fluconazole, and flucytosine (Table 5 and Appendix A).

The results of the quantitative analysis (MIC, MFC) of antifungal activity against pathogenic multidrug yeast strains are shown in Table 5. EmiA was active against the isolates of *C. albicans*, *C. glabrata*, and *C. neoformans* with MIC/MFCs in the range 0.25–4 µg/mL. In agreement with the preliminary results, the lowest activity (MIC = 2 µg/mL) was observed against *C. albicans* 1402. Fluconazole (FZ) showed little or no activity against the whole panel of *Candida* isolates, in accordance with previous findings that these species are resistant to commercial azoles. Interestingly, EmiA exerted strongantifungal activity similar to that of AmpB towards both species of *Cryptococcus* isolates that had intrinsic resistance to caspofungin (CAS) and FZ (Appendix A; Appendix A).

### 3.7. Cytotoxicity and Hemolytic Activity of Emericellipsin A

Taking into account the characteristic specificity of the antimicrobial action of the studied compound with respect to eukaryotic pathogenic organisms, its effectiveness was evaluated for the presence of cytotoxic action on a number of tumor cell lines in comparison to a somatic control line in vitro. The results show that EmiA also has cytotoxic properties towards a wide range of tumor cell lines of various origins. A distinctive feature of EmiA is the contrast pattern of cell growth inhibition, which is expressed in a significant variation in the IC_50_ index. For example, the maximum activity was demonstrated with respect to K-562 (1 μM), while for MCF-7 and B16, the value was more than 10 times higher (11.5 and 16.0 μM, respectively) (Table 6; Appendix A). Most likely, the observed differences are determined by the subtle features of the cell membranes of these lines, as well as the features of their ontogenesis, which provide the initiation of the interaction of this antibiotic from the group of peptaibols with the cell surface.

In general, the antiproliferative activity of emericellipsin A in all variants of the lines was significantly inferior to that of the reference antitumor antibiotic, doxorubicin (by four to 25 times), including the interaction with PFH (60 times). Accordingly, despite the high IC_50_ values for this molecule, fairly significant amounts of antibiotic are required to suppress the cell growth in vitro. The calculated therapeutic index value is sufficiently high due to a low cytotoxicity level for normal cells, which is an undeniable advantage over both pre-scorubicin and most other antitumor compounds, which are components of traditional chemotherapy for the treatment of malignant oncological diseases. In our previous study, EmiA exhibited selective cytotoxic activity against the HepG2 and Hela cell lines (EC_50 =_ 2.8 and <0.5 µM, respectively) [21].

For a more in-depth study of EmiA as a compound with a potentially high therapeutic index and therefore, a less toxic compound with respect to normal human cells, a series of experiments were conducted on its hemolytic activity, which is the ability to destroy the membrane of blood erythrocytes. EmiA demonstrated approximately 12.6% hemolysis at a concentration of 20 μM compared with 100% hemolysis for peptide antibiotic gramicidin S (Figure 4).

We also investigated the possible influence of the peptaibol EmiA on heterochromatin condensation located in the nuclei of human buccal epithelium cells. The data obtained show weak heterochromatin granule formation at active concentrations from 50 µM compared with the control values (Appendix A). This result leads to potential binding of peptaibols to nuclear acids in vitro.

## 4. Discussion

The range of patients at risk of invasive fungal co-infection continues to expand beyond the normal host to encompass patients with acquired immunodeficiency syndrome, those who are immunosuppressed due to therapy for cancer and organ transplantation, and those undergoing major surgical procedures [26,27,28]. It was recently estimated that more than 300 million people have been diagnosed with deep opportunistic mycosis, and *Cryptococcus*, *Candida*, and *Aspergillus* infections account for up to two million deaths annually. Moreover, it has been found that an invasive fungal infection is the main cause of death for SARS patients, accounting for 25–73.7% [27,28,29,30]. Furthermore, *Aspergillus* and *Candida* species could be an important cause of life-threatening infection in COVID-19 patients, especially those with high risk factors [31]. COVID-19 patients with the human immunodeficiency virus (HIV) infection are also susceptible to cryptococcosis, which is predominantly present as meningoencephalitis [28]. Current clinically available drugs have therapeutic limitations due to toxicity, a narrow spectrum of activity, and, more importantly, the consistent rise of fungal species which are intrinsically resistant or develop resistance due to prolonged therapy. The question of fungal resistance to azole drugs is considerably more complex and is currently under evaluation. Examples of both primary and secondary resistance are known for themedically important yeasts and for mold and selected azole antifungals. Both primary and secondary resistance to 5-fluorocytosine are known to occur for strains of *Candida* species. Resistance to the echinocandins is rare, but it does occur for *Aspergillus* spp. and *Cr. neoformans* clinical isolates. Amphotericin B is now utilized when resistance to the azole drugs or the echinocandins is observed, but it is at the high end of the spectrum of toxicity of antifungal drugs, so it is reserved for circumstances such as resistance to other antifungals [1,2,3,4,5].

Fungal peptaibols have attracted intensive attention from both the scientific community and the pharmaceutical industry since the discovery of alamethicin in the late 1960s from the biocontrol fungus *Trichoderma viride* [32,33,34,35]. Many described peptaibols are reported to have an antifungal effect, but most of them possessed activity against plant pathogenic fungi [35,36,37,38,39]. In recent years, there have been a few reports addressing the activity of peptaibols against clinically important fungi [21,40,41]. Bioprospecting for bioactive natural compounds in extremophilic fungi has been reported recently, and the genus *Emericellopsis* has been revealed as a good potential candidate for novel nonribosomal AMPs [10,11,19,21,42,43,44,45]. AMPs derived from novel alkalophilic *Emericellopsis* species may be able to fill the need for an antifungal drug that has little toxicity and can be used as an alternative for fluconazole-resistant fungal strains, especially when echinocandins cannot be used.

In this study, we characterized antifungal nonribosome synthesized AMPs called emericellipsins from the alkalophilic fungus *E. alkalina*, which inhabited soda soils. Primary structural characterization for the novel minor compounds B–E was performed, along with detailed identification of their biological activity towards broad-spectrum fungal and yeast cultures in vitro. One from this complex, EmiA, was identified as the most promising for further studies based on its broad-spectrum antifungal activity and total yield in the crude active concentrate.

In our previous studies, we demonstrated the fungicidal effect on conditionally pathogenic mold and clinical isolates of *Aspergillus* spp. for three major components of emericellipsin’s complex: EmiA, EmiB, and EmiC (previously designated A118/37, A118/35, and A118/36) [19]. The antifungal activity was further confirmed for the MIC range for most previous fungal tests. In the present study, we also showed that the lead compound, EmiA, is highly effective towards *A. terreus* strains that have primary resistance to amphotericin B. In contrast, the two minor components of complexes EmiD and EmiE showed no antimicrobial activity in our initial tests. The antifungal activity decreased in the peptaibol line (EmiA→EmiE).

These results are quite interesting since all novel emericellipsins (B–E) contain minor structural differences from the reference structure of EmiA. In most cases, they are represented by a single amino acid substitution. It is important to note that nonribosomal peptides belonging to the peptaibols typically have a helical spatial conformation [46,47,48]. However, this orientation may be α-helical with an additional 3_(10)-_helix [49,50]. We discovered here that emericellipsins B–E are close to form A, but their antifungal activity was decreased, as indicated above. This could be associated with local structural displacements, followed by some changes in helical conformation. It was shown that it is critical for peptaibols to integrate into fungal plasma membranes and realize the pore-forming mechanism of action [50].

There is a wide structural diversity among peptaibols from *Trichoderma* spp., many of which are found to have single amino acid substitutions [51]. In some cases, peptaibols from *Trichoderma* are isolated as groups of homologous peptides that have certain variability in their amino acid sequence, which influences the quantitative level of biological activity, particularly cytotoxicity as a rule [52,53,54]. Semi-synthetic analogues based on peptaibol trichogin isolated from *Trichoderma* sp. were designed and synthesized to increase the water solubility and antifungal activity against an economically important fungal plant pathogen, *Botrytis cinerea*. These molecules have single amino acid substitutions, predominantly Gly/Lys located inside a helical structure [55]. It was shown that this modification increased peptide amphiphilicity and did not lead to critical changes of 3D orientation. Concerning functional aspects, most of the mutant peptaibols could improve the inhibition effect towards *B. cinerea* [54].

The activity of EmiA was investigated in more detail against an extended panel of multidrug-resistant clinical isolatesof *Candida* and *Cryptococcus*. The results revealed resistance to fluconazole and intraconazole and low susceptibility to caspofungin. Among the clinically approved antifungal drugs, only amphotericin B displays a comparable activity profile towards the drug-resistant *C. neoformans* and *C. laurentii* isolates obtained from HIV-positive patients with cryptococcal meningitis [28,31,55,56].

In our experiments, very low cytotoxicity and hematoxicity were observed. We discovered cytotoxic action of EmiA towards a spectrum of tumor cell lines in vitro. Thus, we determined the IC_50_ values for a wide range of active concentrations (1‒16 µM) that could potentially result in different resistance among cell lines to the antibiotic action. Moreover, the comparison of dose-dependent curves of EmiA and doxorubicin could reveal that they are different overall. As a result, the application of doxorubicin to cells at a low concentration led to high mortality, whereas incubation with EmiA was strictly dose-dependent [21]. Unusually, however, EmiA was able to act against heterochromatin condensation in human buccal epithelium cells, which was achieved at a dose of 50 µM and was concentration-dependent. This effect was firstly described for nonribosomal antimicrobial peptides from the peptaibol group and requires more detailed consideration [49,50]. In a complementary toxicity study, the hemolysis profiles of the peptide and GramS were compared using the absorbance values of the released hemoglobin, as shown in Figure 4. Emericellipsin A revealed negligible hemolytic activity at concentrations of 0–20 μM, making it a low-toxicity compound with respect to normal human cells, but with a potentially high therapeutic index. Our results were also compared with the toxic levels of AmpB that could induce hemolytic activity at the tested concentration levels [57].

## 5. Conclusions

In conclusion, this study is the first to elucidate the antifungal activity of an AMP derived from the alkalophillic fungi *E. alkalina* against multidrug-resistant clinical fungal pathogens. The major component, EmiA, demonstrated a broad-spectrum fungal action against clinical yeast and filamentous fungi, low toxicity, and no hemolytic activity with respect to normal human cells. These findings might provide insights into the possible therapeutic application of EmiA as an antifungal agent to treat invasive fungal infections, such as cryptococcosis, candidiasis, and aspergillosis.

## Figures and Tables

**Figure 1 jof-07-00153-f001:**
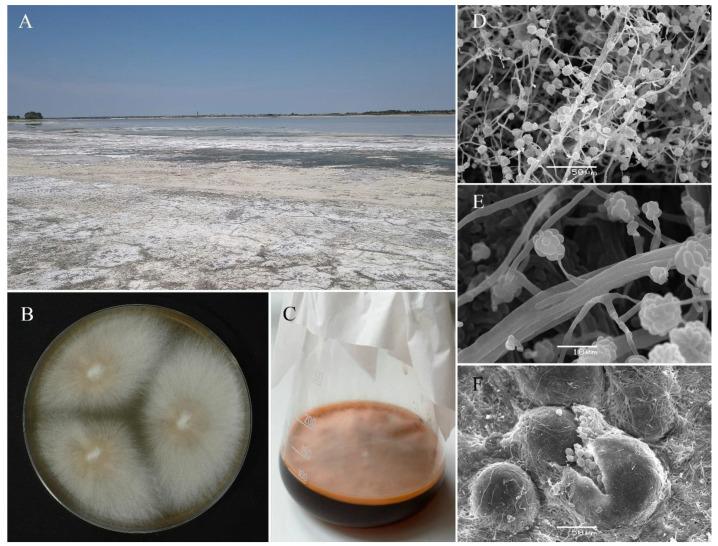
*Emericellopsis alkaline* in situ and in vitro. (**A**) The edge of the soda lake, typical habitat for *E. alkalina* (Lake Tanatar-2, Altai area, Russia); (**B**) growth of *E. alkalina* F-1428 on the alkaline agar medium and (**C**) at the liquid alkaline medium; (**D**,**E**) *E. alkalina* co-nidiophores; and (**F**) fruiting body, SEM.

**Figure 2 jof-07-00153-f002:**
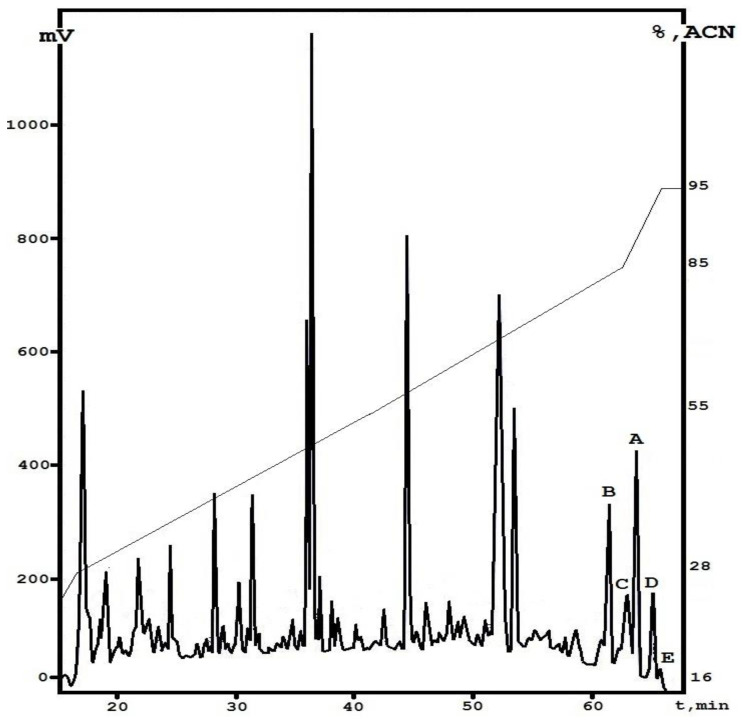
Fractionation of the *E. alkalina* concentrate by analytical reversed-phase HPLC. Detection of absorbance was monitored at 214 nm. The target components are marked by letters A–E.

**Figure 3 jof-07-00153-f003:**
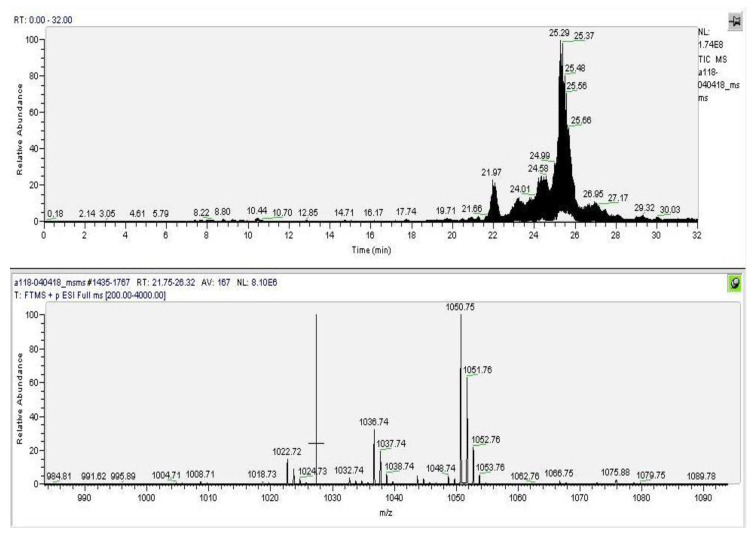
Electro spray ionization–mass spectrometry (ESI–MS) profile of the mixture fraction containing components A, B, C, D, and E. Designations: Ion current (upper panel), m/z values, Da (down current).

**Figure 4 jof-07-00153-f004:**
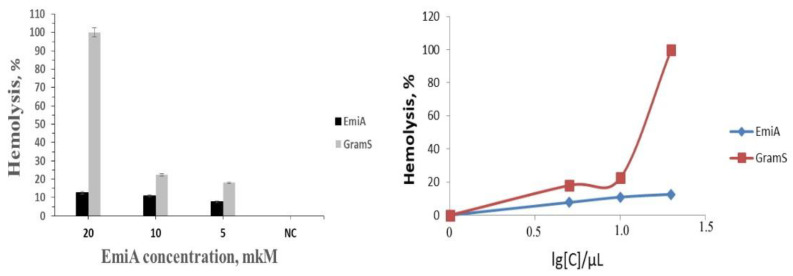
Hemolytic activity of EmiA and gramicidin S (GramS) on human erythrocytes. Designations: NC: Negative control (0.9% solution of NaCl); the assay was repeated in triplicate, and the percentage of hemolysis is expressed as the mean ± SD.

**Table 1 jof-07-00153-t001:** *E. alkalina* strains involved in the study. Antifungal activity in the agar diffusion assay and detection of emericellipsins in crude extracts.

No. of Strains, No. of VKM, VKPM, CBS Collections	Origin (Isolation Place, Soil’s Saltification Type)	Activity of Crude Peptaibols Extracts, Zone of Inhibition, mm		Total Content of EmiA	Presence of Homologues B–E
		*A. niger* ATCC 16404	*C. albicans* ATCC 14053		
E101 T =VKM F-4108; =CBS 127350	Tanatar-II lake *, soda	10 ± 0.2	0	0.61%	d.
A118 =VKPM F1428	Zheltir’ lake *, soda-chloride	26 ± 0.1	28 ± 0.6	1.2%	d.
A103	Mirabilit lake *, soda-chloride-sulfate	18 ± 0.1	15 ± 0.1	0.9%	d.
A114 =VKM FW-1473	Solyonoe lake *, chloride	25 ± 0.6	12 ± 0.6	0.05%	d.
A120	Bezimyannoe lake *, soda	12 ± 0.1	0	0.07%	n.d.
A121 VKM FW-1475	Tanatar-II lake *, soda	13 ± 0.4	9 ± 0.1	0.1%	n.d.
M20 =VKM FW-3040; =CBS 120044	Zheltir’ lake *, soda-chloride	26 ± 0.1	28 ± 0.6	1.07%	d.
3KS17-13	Malinovoye lake *, chloride	7 ± 0.2	0	0.63%	n.d.
5KS17-3	Tanatar-II lake *, soda	12 ± 0.2	0	0.42%	n.d.
6KS17-1	Tanatar-II lake *, soda	6 ± 0.2	0	0.42%	n.d.
7KS17-1	Tanatar-II lake *, soda	12 ± 0.2	0	0.42%	n.d.
8KS17-1	Tanatar-II lake *, soda	11 ± 0.2	0	n.d.	d.
9KS17-3	Tanatar-II lake *, soda	14 ± 0.2	8 ± 0.2	0.54%	n.d.
10KS17-1	Tanatar-II lake *, soda	15 ± 0.2	0	n.d.	d.
14KS17-1	Tanatar I lake *, soda	9 ± 0.2	12 ± 0.2	0.42%	d.
A126 =VKM FW-1472	Transbaikal, Nuhe-Nur lake, soda	12 ± 0.1	0	0.82%	n.d.
A128	Transbaikal, Sulfatnoe lake, sulfate-soda	14 ± 0.1	10 ± 0.1	0.12%	d.

T: Type strain; * Kulunda steppe, Altai; d.: Detected; n.d.: Not detected; the largest-volumepeptide’s complex producer marked by red color.

**Table 2 jof-07-00153-t002:** Initial structural analysis of the emericellipsins A–E.

Peptaibol	M+H, Da	Amino Acid Sequence
EmiA	1050.69	Methyldecanoyl-MePro-AHMOD-Ala-Aib-Ile-Iva-βAla-Alaol-Glyol
EmiB	1036.77	Methyldecanoyl-MePro-AHMOD-Ala-Aib-Ile-Aib-βAla-Alaol-Glyol
EmiC	1066.76	Methyldecanoyl-MePro-AHMOD-Ser-Aib-Ile-Iva-βAla-Alaol-Glyol
EmiD	1052.76	Methyldecanoyl-MePro-AHMOD-Ala-Iva-Ile-Iva-βAla-Alaol-Glyol
EmiE	1079.75	Methyldecanoyl-MePro-AHMOD-Ala-Aib-Ile-Iva-βAla-Aib-Glyol

All the substitutions according to the EmiA sequence are in red. The reference structure of EmiA is framed.

**Table 3 jof-07-00153-t003:** Biological comparison data of crude peptaibols extract and pure emericellipsins on clinical and opportunistic mold fungi isolates (40 µg/disc).

Strain	Zone of Inhibition, mm						
	Compound						
	EmiA	EmiB	EmiC	EmiD	EmiE	Crude Peptaibols Complex	AmpB
*A. niger* ATCC 16404	25 ± 0.21	10 ± 0.13	9 ± 0.31	0	0	20 ± 0.24	0
*A. niger* 1133 m *	18 ± 0.27	11 ± 0.170	0	0	0	12 ± 0.21	10 ± 0.14
*A. fumigatus* VKM F-37	20 ± 0.3	0	0	0	0	12 ± 0.2	12 ± 0.23
*A. fumigatus* 390 m *	20 ± 0.40	0	0	0	0	10 ± 0.21	15 ± 0.51
*A. terreus* 3 K	12 ± 0.25	0	0	0	0	10 ± 0.31	0
*A. terreus* 497 *	14 ± 0.28	0	0	0	0	0	0

* Pathogenic clinical *Aspergillus* isolates.

**Table 4 jof-07-00153-t004:** Minimal inhibitory concentrations (MICs) of the emericellipsins antimicrobial peptides (AMPs) against different clinical and opportunistic fungi.

Strain	Minimal Inhibitory Concentration (MIC) (µg/mL)					
	Compound					
	EmiA	EmiB	EmiC	EmiD	EmiE	AmpB
*A. niger* ATCC 16404	4	8	32	˃64	˃64	1
*A. niger* 1133 m *	4	8	32	˃64	˃64	1
*A. terreus* 3 K	0.5	4	8	16	32	˃64
*A. terreus* 497 *	1	4	8	16	64	˃64
*A. fumigatus* VKM F-37	2	8	16	32	˃64	1
*A. fumigatus* 390 m *	2	8	16	32	˃64	1

* Pathogenic clinical *Aspergillus* isolates.

**Table 5 jof-07-00153-t005:** MIC and minimal fungicidal concentration (MFC) of emericellipsin A against clinical yeast isolates.

Strain	Minimal Inhibitory/Fungicidal Concentration (MIC/MFC) (µg/mL)						
	Compound						
	EmiA		AmpB		CAS		FZ
	MIC	MFC	MIC	MFC	MIC	MFC	
*Candida albicans* 1402	2	4	0.5	0.5	0.06	0.5	64
*C. glabrata* 1402	0.25	0.5	1	1	0.06	0.5	128
*C. krusei* 1447	0.5	2	2	4	0.25	1	R
*C. tropicalis* 156	1	2	0.5	1	0.06	0.5	64
*C. parapsilosis* 571	1	1	1	1	1	1	128
*Cryptococcus neoformans* 297	0.5	0.5	0.5	1	>64	n.d.	16
*C. laurentii* 325m	0.5	0.5	0.25	0.5	16	n.d.	32

FZ: Fluconazole; CAS: Caspofungin; AmpB: Amphotericin B; EmiA: Emericellipsin A.

**Table 6 jof-07-00153-t006:** IC_50_ values of emericellipsin A compared with a control antitumor antibiotic (doxorubicin).

Compound	IC_50,_ _µM_					
	HCT-116	MCF-7	HPF	K-562	B16	MDA-MB-231
EmiA	2.30 ± 0.30	11.0 ± 1.20	11.5 ± 1.50	1.00 ± 0.14	16.0 ± 2.08	8.00 ± 1.04
Doxorubicin	0.20 ± 0.03	0.50 ± 0.06	0.20 ± 0.03	0.25 ± 0.03	0.60 ± 0.07	0.80 ± 0.09

## Data Availability

Not applicable.

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
