# Peer review of "The Emericellipsins A–E from an Alkalophilic Fungus Emericellopsis alkalina Show Potent Activity against Multidrug-Resistant Pathogenic Fungi"

_jof, 2021, doi:10.3390/jof7020153_

Round 1

Reviewer 1 Report

The authors emphasize the need for the use of the antimicrobial peptides considered primising alternative to antifungal treatment during the infections caused by pathogenic fungi MDR.

The theme is interesting for the potential low toxicity associated with the use of  Emy A,

but it is necessary to specify :

the time growing conditions and if it is  cause of different effects on Emericellopsis production.

Please answer some other questions:

if you tried to obtain a sintetic production of Emy A and if it is more efficacy; 

if you studied the possibility of sinergic effect   associated with the use of antimicrobial peptides and fluconazole.

Author Response

Dear Reviewer! Thank you very much for reviewing our paper! 

the time growing conditions and if it is the cause of different effects on Emericellopsis production.
Thank you for pointing this moment! Our previous results showed that the studied F - 1428 producer of E. alkalina cultivated in the liquid alkaline medium had monophasic exponential growth from inoculation until day 7. At the end of the exponential and beginning of the stationary phase (day 7–10), the maximal total dry weight biomass and (day 10 – 14) EmiA synthesis were observed. After 14 days, autolysis began due to the exhausting of sucroses from the medium, which was reflected as an insignificant reduction of biomass at the end of the experiment. 
We added this information in the “2.2. Cultivation of the Fungi and Extraction of Emericellipsins A‒E”, Line 102.

Please answer some other questions:
if you tried to obtain a sintetic production of Emy A and if it is more efficacy; 
We have, unfortunately, not yet succeeded in chemical synthesis of this compound, but we hope to have such a result soon.
if you studied the possibility of synergic effect associated with the use of antimicrobial peptides and fluconazole.
Nobody raised the question about the synergic effect of fluconazole and our peptides because of azole resistance for the most investigated clinical fungi isolates tested in our experiments.  Either way, it requires additional detailed experiments that will be performed in further study.

Reviewer 2 Report

Overall I enjoyed very much reading this manuscript. Nonetheless, I feel that the manuscript should be improved prior to publication:

English needs revision by a native speaker. These are just some examples of sentences that I feel need revisions:

- Lanes 18/19: “We previously reported that this strain gave the emericellipsin A (EmiA) with strong antifungal…”

- Lanes 35/36: “…there is a real and significant threat to human health as a result of by incidence of infections…”

-  use the dot as the decimal point (eg. Table 1)

- The authors state that “There is a real and significant threat to human health as a result of by incidence of infections caused by pathogenic fungi..”. Can the authors quantify this incidence? Usually reports describe nosocomial bacterial infections, but not fungal infections. It would be nice to see some numbers.

- Can the authors indicate some references that support the sentence: “The majority of natural AMPs are of mammalian origin (~75%), followed by plant (~13%) and bacterial origin (~10%). Only 1% of the currently known and 45 studied AMPs are from fungi.”

- the web address for the Norine database does not seem to be correct;

- in lanes 221/222 I think that Fig.1 is not well addressed;

- what do the authors mean by “characterization at the structural and functional levels”? (lanes 250/251);

- “caspofungin, itraconazole, voriconazole, and amphotericin B” are not antibiotics.

- in general, figure legends should be more complete. For example, figure S3 legend should indicate what are PFH-EmiA, B16-EmiA, etc.; in figure S4, what are donors? In figure 4, explain to what the “density” corresponds.

Author Response

Overall I enjoyed very much reading this manuscript. Nonetheless, I feel that the manuscript should be improved prior to publication:

Dear Reviewer! Thank you very much for giving us such a positive review and comments, which make our manuscript more in line with the requirements of the Journal of Fungi. We have revised our manuscript according to your comments and established a new submission. Our responses to your opinion are as follows:

English needs revision by a native speaker. These are just some examples of sentences that I feel need revisions:

- Lanes 18/19: “We previously reported that this strain gave the emericellipsin A (EmiA) with strong antifungal…”

- Lanes 35/36: “…there is a real and significant threat to human health as a result of by incidence of infections…”

-  use the dot as the decimal point (eg. Table 1)

 Thank you so much, that you noted these grammar errors in the text.  This manuscript was proofread by the MDPI service (English editing ID: English-26887), the certificate was uploaded. Also, I have modified table1.

- The authors state that “There is a real and significant threat to human health as a result of by incidence of infections caused by pathogenic fungi.”. Can the authors quantify this incidence? Usually, reports describe nosocomial bacterial infections, but not fungal infections. It would be nice to see some numbers.

Thank you very much for your comment. When we were writing the manuscript, we were guided by the following information from web site http://www.life-worldwide.org/:

For example:

(http://www.life-worldwide.org/fungal-diseases/candidaemia-and-invasive-candidiasis)

Candidemia occurs at a population rate of 2-11/100,000, so ~350,000 cases are predicted worldwide every year. The numbers rose in the US by 52% between 2000 and 2005. Rates in India and Brazil are much higher, so the overall estimate could be greater. 

(http://www.life-worldwide.org/fungal-diseases/invasive-aspergillosis)

Over 30 million people are at risk of invasive aspergillosis each year because of corticosteroid or other therapies, and over 300,000 patients develop it annually. Worldwide, at least 125,000 of these cases are in COPD.

There is also some recent information about invasive aspergillosis diseases statistics in references 1-3:

  1. Antinori, L.M.; Sollima, S.; Galli, M.; Corbellino, M. Candidemia and invasive candidiasis in adults: A narrative review. J. Intern. Med. 2016, 34, 21–28, doi:10.1016/j.ejim.2016.06.029.
  2. Chastain, D.B.; Henao-Martınez, A.F.; Franco-Paredes, C. Opportunistic invasive mycoses in AIDS: Cryptococcosis, histoplasmosis, coccidiodomycosis, and talaromycosis. Curr Infect DisRep. 2017, 19, 1-9, doi:10.1007/s11908-017-0592-7.
  3. Chaturvedi, V.; Bouchara, J.P.; Hagen, F.; Alastruey-Izquierdo, A.; Badali, H.; Bocca, A.L.; et.al. Eighty years of mycopathologia: a retrospective analysis of progress made in understanding human and animal fungal pathogens. Mycopathologia 2018, 183, 859–877, doi:10.1007/s11046-018-0306-1.

The web address has been included in the introduction section and highlighted in yellow color.

- Can the authors indicate some references that support the sentence: “The majority of natural AMPs are of mammalian origin (~75%), followed by the plant (~13%) and bacterial origin (~10%). Only 1% of the currently known and 45 studied AMPs are from fungi.”

Thank you for your comment. Meyer et all, 2018 reported thatThe majority of natural AMPs are of mammalian origin (~75%), followed by a plant (~13%) and bacterial origin (~10%). Only 1% of the currently known and 45 studied AMPs are from fungi”

 Meyer, V.; Jung, S. Antifungal Peptides of the AFP Family Revisited: Are These Cannibal Toxins? Microorganisms. 2018, 6, 1-15, DOI: 10.3390/microorganisms6020050.

The reference has been added.

- the web address for the Norine database does not seem to be correct;

Thank you. We rewrote it.

- in lanes 221/222 I think that Fig.1 is not well addressed;

Thank you so much for a recommendation that Figure 1 is better referenced elsewhere. We deleted the reference in line 222, and added it in section 2.1 (line 88 – 91) as follows: “All strains were isolated from soils adjacent to soda or saline lakes (Fig. 1 A) and all of them showed the growth ability at рН 10.5 media: alkaline agar (Fig. 1 B) and liquid alkaline medium (Fig. 1 C). The scanning electron microscopy (SEM) was used for morphological study of the strains (Fig. 1 D-F)”

- what do the authors mean by “characterization at the structural and functional levels”? (lanes 250/251);

We rewrote its sentence as follows: “The other dominant peaks were collected manually to provide initial structural analysis”

- “caspofungin, itraconazole, voriconazole, and amphotericin B” are not antibiotics.

Thank you for pointing that out! We have improved the formulation. Here we corrected ‘antibiotic’ to antifungal drugs. It is corrected to “antifungal drugs” throughout the manuscript.

- in general, figure legends should be more complete. For example, figure S3 legend should indicate what are PFH-EmiA, B16-EmiA, etc.; in figure S4, what are donors? In figure 4, explain to what the “density” corresponds.

We included all necessary information in Fig4 and S 4 legends and material and methods section